# Neural Symbolic Machines:
# Learning Semantic Parsers on Freebase
# with Weak Supervision

## Abstract

Harnessing the statistical power of neural networks to perform language understanding and symbolic reasoning is difficult, when it requires executing efficient discrete operations against a large knowledge-base. In this work, we introduce a Neural Symbolic Machine, which contains (a) a neural "programmer", i.e., a sequence-to-sequence model that maps language utterances to programs and utilizes a *key-variable memory* to handle compositionality (b) a symbolic "computer", i.e., a Lisp interpreter that performs program execution, and helps find good programs by pruning the search space. We apply REINFORCE to directly optimize the task reward of this structured prediction problem. To train with weak supervision and improve the stability of REINFORCE we augment it with an *iterative maximum-likelihood* process. NSM outperforms state-of-the-art on the WEBQUESTIONSSP dataset when trained from question-answer pairs only, without requiring any feature engineering or domain-specific knowledge.

## 1 Introduction

Deep neural networks have achieved impressive performance in supervised classification and structured prediction tasks such as speech recognition (Hinton et al., 2012), machine translation (Bahdanau et al., 2014; Wu et al., 2016) and more. However, training neural networks for semantic parsing (Zelle and Mooney, 1996; Zettlemoyer and Collins, 2005; Liang et al., 2011) or program induction, where language is mapped to a symbolic representation that is executed by a com-

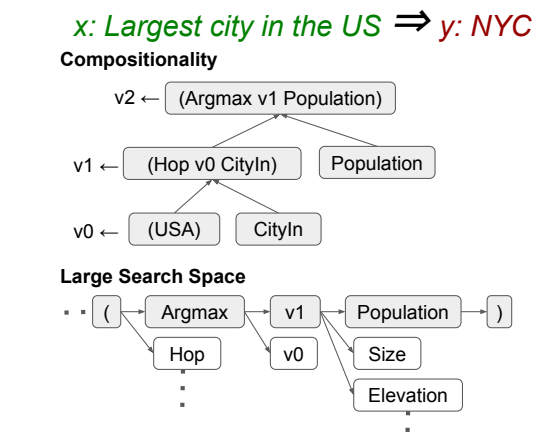

*x: Largest city in the US* ⟹ *y: NYC*

Figure 1: The main challenges of training a semantic parser from weak supervision: (a) *compositionality*: we use variables $(v_0, v_1, v_2)$ to store execution results of intermediate generated programs. (b) *search*: we prune the search space and augment REINFORCE with pseudo-gold programs.

puter, through weak supervision remains challenging. This is because the model must interact with a computer through non-differentiable operations at training time.

In semantic parsing, recent work handled this (Dong and Lapata, 2016; Jia and Liang, 2016) by training from manually annotated programs and avoiding program execution at training time. However, annotating programs is known to be expensive and scales poorly. In program induction, attempts to address this problem (Graves et al., 2014; Reed and de Freitas, 2016; Kaiser and Sutskever, 2015; Graves et al., 2016b; Andreas et al., 2016) either utilized low-level memory (Zaremba and Sutskever, 2015), or required memory to be differentiable (Neelakantan et al., 2015) so that the model can be trained with backpropagation. This makes it difficult to use the efficient discrete operations and memory of a traditional computer, and limited the application to synthetic or small knowledge bases.

In this paper, we propose to utilize the mem-

ory and discrete operations of a traditional computer in a novel Manager-Programmer-Computer (MPC) framework for neural program induction, which integrates three components:

1. A **"manager"** that provides weak supervision (e.g., 'NYC' in Figure 1) through a reward indicating how well a task is accomplished. Unlike full supervision, weak supervision is easy to obtain at scale (Section 3.1).

2. A **"programmer"** that takes natural language as input and generates a program that is a sequence of tokens (Figure 2). The programmer learns from the reward and must overcome the hard search problem of finding correct programs (Section 2.2).

3. A **"computer"** that executes programs in a high level programming language. Its non-differentiable memory enables *abstract*, *scalable* and *precise* operations, but makes training more challenging. It provides a friendly *neural computer interface* by detecting and eliminating invalid choices (Section 2.1).

Within this framework, we introduce the Neural Symbolic Machine (NSM) and apply it to semantic parsing. NSM contains a sequence-to-sequence (seq2seq) "programmer" (Sutskever et al., 2014) and a symbolic non-differentiable Lisp interpreter ("computer") that executes programs against a large knowledge-base (KB).

Our technical contribution in this work is threefold. First, to support language *compositionality*, we augment the standard seq2seq model with a *key-variable memory* to save and reuse intermediate execution results (Figure 1). This is a novel application of pointer networks (Vinyals et al., 2015) to compositional semantics.

Second, to alleviate the search problem of finding correct programs when training from question-answer pairs (Figure 1), we use the computer to execute partial programs and prune the programmer's search space by checking the syntax and semantics of generated programs. This generalized past work on training from weak supervision (Liang et al., 2011; Berant et al., 2013) by using the semantic denotations for structural search.

Third, to train from weak supervision and directly maximize the expected reward we turn to the REINFORCE (Williams, 1992) algorithm. Since learning from scratch is difficult for REINFORCE, we combine it with an iterative maximum likelihood (ML) process, where beam search is used to find pseudo-gold programs, which are then used to augment the objective of REINFORCE.

On the WEBQUESTIONSSP dataset (Yih et al., 2016), NSM achieves new state-of-the-art results with weak supervision, significantly closing the gap between weak and full supervision for this task. Unlike prior work, it is trained end-to-end, and does not require feature engineering or domain-specific knowledge.

## 2 Neural Symbolic Machines

We now introduce NSM by first describe the "computer", a non-differentiable Lisp interpreter that executes programs against a large KB and provides code assistance (Section 2.1). We then propose a seq2seq model ("programmer") that supports compositionality using a key-variable memory to save and reuse intermediate results (Section 2.2). Finally, we describe a training procedure that is based on REINFORCE, but is augmented with pseudo-gold programs found by an iterative ML training procedure (Section 2.3).

Before diving into details, we define the *semantic parsing* task: given a knowledge base $\mathbb{K}$, and a question $x = (w_1, w_2, ..., w_m)$, produce a program or logical form $z$ that when executed against $\mathbb{K}$ generates the right answer $y$. Let $\mathcal{E}$ denote a set of entities (e.g., ABELINCOLN),[1] and let $\mathcal{P}$ denote a set of properties (e.g., PLACEOFBIRTH). A knowledge base $\mathbb{K}$ is a set of assertions or triples $(e_1, p, e_2) \in \mathcal{E} \times \mathcal{P} \times \mathcal{E}$, such as (ABELINCOLN, PLACEOFBIRTH, HODGENVILLE).

### 2.1 Computer: Lisp Interpreter with Code Assistance

Semantic parsing typically requires using a set of operations to query the knowledge base and process the results. Operations learned with neural networks such as addition and sorting do not perfectly generalize to inputs that are larger than the ones observed in the training data (Graves et al., 2014; Reed and de Freitas, 2016). In contrast, operations implemented in high level programming languages are *abstract*, *scalable*, and *precise*, thus generalizes perfectly to inputs of arbitrary size. Based on this observation, we implement operations necessary for semantic parsing with an ordinary programming language instead of trying to learn them with a neural network.

---

[1] We also consider numbers (e.g., "1.33") and date-times (e.g., "1999-1-1") as entities.

| | |
|---|---|
| $(\ Hop\ v\ p\ ) \Rightarrow \{e_2 | e_1 \in v, (e_1, p, e_2) \in \mathbb{K}\}$ | |
| $(\ ArgMax\ v\ p\ ) \Rightarrow \{e_1 | e_1 \in v, \exists e_2 \in \mathcal{E} : (e_1, p, e_2) \in \mathbb{K}, \forall e : (e_1, p, e) \in \mathbb{K}, e_2 \geq e\}$ | |
| $(\ ArgMin\ v\ p\ ) \Rightarrow \{e_1 | e_1 \in v, \exists e_2 \in \mathcal{E} : (e_1, p, e_2) \in \mathbb{K}, \forall e : (e_1, p, e) \in \mathbb{K}, e_2 \leq e\}$ | |
| $(\ Filter\ v_1\ v_2\ p\ ) \Rightarrow \{e_1 | e_1 \in v_1, \exists e_2 \in v_2 : (e_1, p, e_2) \in \mathbb{K}\}$ | |

Table 1: Interpreter functions. $v$ represents a variable, $p$ a property in Freebase. $\geq$ and $\leq$ are defined on numbers and dates.

We adopt a Lisp interpreter as the "computer". A program $C$ is a list of expressions $(c_1...c_N)$, where each expression is either a special token "RETURN" indicating the end of the program, or a list of tokens enclosed by parentheses "( $F\ A_0\ ...\ A_K$ )". $F$ is one of the functions in Table 1, which takes as input $K$ arguments of specific types. Table 1 defines the semantics of each function and the types of arguments (either a property $p$ or a variable $v$). When functions are executed, they return an entity list that is the expression's denotation in $\mathbb{K}$, and save it to a new variable.

By introducing variables that save the intermediate results of execution, the program naturally models *language compositionality* and describes from left to right a bottom-up derivation of the full meaning of the natural language input, which is convenient in a seq2seq model (Figure 1). This is reminiscent of the floating parser (Wang et al., 2015; Pasupat and Liang, 2015), where a derivation tree that is not grounded in the input is incrementally constructed.

The set of programs defined by our functions is equivalent to the subset of $\lambda$-calculus presented in (Yih et al., 2015). We did not use full Lisp programming language here, because constructs like control flow and loops are unnecessary for most current semantic parsing tasks, and it is simple to add more functions to the model when necessary.

To create a friendly *neural computer interface*, the interpreter provides code assistance to the programmer by producing a list of valid tokens at each step. First, a valid token should not cause a syntax error: e.g., if the previous token is "(", the next token must be a function name, and if the previous token is "*Hop*", the next token must be a variable. More importantly, a valid token should not cause a semantic (run-time) error: this is detected using the denotation saved in the variables. For example, if the previously generated tokens were "( *Hop* $v$", the next available token is restricted to properties $\{p \mid \exists e, e' : e \in v, (e, p, e') \in \mathbb{K}\}$ that are reachable from entities in $v$ in the KB graph. These checks are enabled by the variables and can

be derived from the definition of the functions in Table 1. With this code assistance, the interpreter prunes the "programmer"'s search space by orders of magnitude, and enables learning from weak supervision on a large KB.

## 2.2 Programmer: Seq2seq Model with Key-Variable Memory

Given the "computer", the "programmer" needs to map natural language into a program, which is a sequence of tokens that references operations and values in the "computer". We base our programmer on a standard seq2seq model with attention, but extend it with a key-variable memory that allows the model to learn to represent and refer to program variables (Figure 2).

Sequence-to-sequence models consist of two RNNs, an encoder and a decoder. We used a 1-layer GRU (Cho et al., 2014) for both the encoder and decoder. Given a sequence of words $w_1, w_2...w_m$, each word $w_t$ is mapped to an embedding $q_t$ (embedding details are Section 3). Then, the encoder reads these embeddings and updates its hidden state step by step using $h_{t+1} = GRU(h_t, q_t, \theta_{Encoder})$, where $\theta_{Encoder}$ are the GRU parameters. The decoder updates its hidden states $u_t$ by $u_{t+1} = GRU(u_t, c_{t-1}, \theta_{Decoder})$, where $c_{t-1}$ is the embedding of last step's output token $a_{t-1}$, and $\theta_{Decoder}$ are the GRU parameters. The last hidden state of the encoder $h_T$ is used as the decoder's initial state. We also adopt a dot-product attention similar to Dong and Lapata (2016). The tokens of the program $a_1, a_2...a_n$ are generated one by one using a softmax over the vocabulary of valid tokens at each step, as provided by the "computer" (Section 2.1).

To achieve compositionality, the decoder must learn to represent and refer to intermediate variables whose value was saved in the "computer" after execution. Therefore, we augment the model with a **key-variable memory**, where each entry has two components: a continuous embedding key $v_i$, and a corresponding variable token $R_i$ referencing the value in the "computer" (see Figure 2).

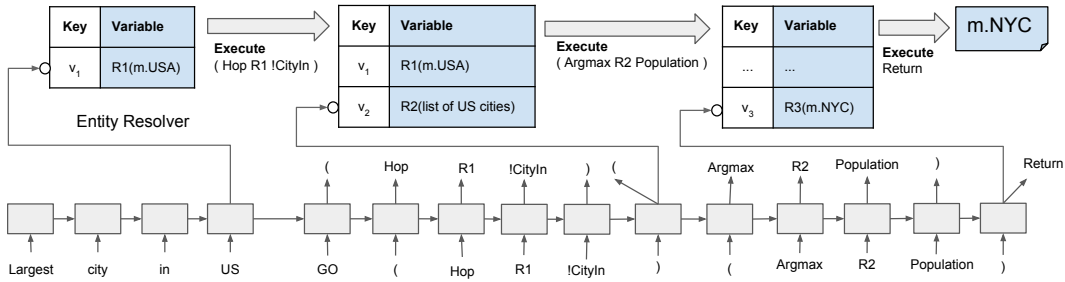

Figure 2: Semantic Parsing with NSM. The key embeddings of the key-variable memory are the output of the sequence model at certain encoding or decoding steps. For illustration purposes, we also show the values of the variables in parentheses, but the sequence model never sees these values, and only references them with the name of the variable ("R1"). A special token "GO" indicates the start of decoding, and "RETURN" indicates the end of decoding.

During encoding, we use an entity linker to link text spans (e.g., *"US"*) to KB entities. For each linked entity we add a memory entry where the key is the average of GRU hidden states over the entity span, and the variable token (*R1*) is the name of a variable in the computer holding the linked entity (*m.USA*) as its value. During decoding, when a full expression is generated (i.e., the decoder reads ")"), it gets executed, and the result is stored as the value of a new variable in the "computer". This variable is keyed by the GRU hidden state at that step. Every time a new variable is added to memory, the variable token is added to the output vocabulary of the decoder. The final answer returned by the "programmer" is the value of the last computed variable.

Note that, similar to pointer networks (Vinyals et al., 2015), the key embeddings for variables are dynamically generated for each example. During training, the model learns to represent variables by backpropagating gradients from a time step where a variable is selected by the decoder, through the key-variable memory, to an earlier time step when the key embedding was computed. Thus, the encoder/decoder learns to generate representations for variables such that they can be used at the right time to construct the correct program.

While the key embeddings are differentiable, the values referenced by the variables (lists of entities), stored in the "computer", are symbolic and non-differentiable. This distinguishes the key-variable memory from other memory-augmented neural networks that use continuous differentiable embeddings as the value of each memory entry (Weston et al., 2014; Graves et al., 2016a).

## 2.3 Training NSM with Weak Supervision

NSM executes non-differentiable operations against a KB, and thus end-to-end backpropa-

gation is not possible. Therefore, we base our training procedure on REINFORCE (Williams, 1992; Norouzi et al., 2016).

When the reward signal is sparse and the search space is large, it is common to utilize some full supervision to pre-train REINFORCE (Silver et al., 2016). Since we assume only weak supervision, we suggest an iterative ML procedure for finding pseudo-gold programs that will bootstrap REINFORCE.

**REINFORCE** We can formulate training as a reinforcement learning problem: given a question $x$, the state, action and reward at each time step $t \in \{0, 1, ..., T\}$ are $(s_t, a_t, r_t)$. Since the environment is deterministic, the state is defined by the question $x$ and the action sequence: $s_t = (x, a_{0:t-1})$, where $a_{0:t-1} = (a_0, ..., a_{t-1})$ is the history of actions at time $t$. A valid action at time $t$ is $a_t \in A(s_t)$, where $A(s_t)$ is the set of valid tokens given by the "computer". Since each action corresponds to a token, the full history $a_{0:T}$ corresponds to a program. The reward $r_t = I[t = T] \cdot F_1(x, a_{0:T})$ is non-zero at the last step of decoding, and is the $F_1$ score computed comparing the gold answer and the answer generated by executing the program $a_{0:T}$. Thus, the reward of a program or action sequence is

$$R(x, a_{0:T}) = \sum_t r_t = F_1(x, a_{0:T}).$$

The agent's decision making is characterized by a policy, $\pi_\theta(s, a) = P_\theta(a_t = a | x, a_{0:t-1})$, where $\theta$ are the model parameters. Since the environment is deterministic, the probability of an action sequence $a_{0:T}$ is

$$P_\theta(a_{0:T}|x) = \prod_t P_\theta(a_t \mid x, a_{0:t-1}).$$

We can define our objective to be the expected reward and use policy gradient methods such as

REINFORCE for training. The objective and gradient are:

$$J^{RL}(\theta) = \sum_x \mathbb{E}_{P_\theta(a_{0:T}|x)}[R(x, a_{0:T})],$$

$$\nabla_\theta J^{RL}(\theta) = \sum_x \sum_{a_{0:T}} P_\theta(a_{0:T} \mid x) \cdot [R(x, a_{0:T}) -$$

$$B(x)] \cdot \nabla_\theta \log P_\theta(a_{0:T} \mid x),$$

where $B(x) = \sum_{a_{0:T}} P_\theta(a_{0:T} \mid x) R(x, a_{0:T})$ is a baseline that reduces the variance of the gradient estimation without introducing bias.

While REINFORCE assumes a stochastic policy, we rely on beam search for gradient estimation. Thus, in contrast with common practice of approximating the gradient by sampling from the model, we use the top-$k$ action sequences (programs) in the beam with normalized probabilities. This allows training to focus on sequences with high probability, which are on the decision boundaries, and reduces the variance of the gradient.

Empirically (and in line with prior work), REINFORCE converged slowly and often got stuck in local optima (see Section 3). The difficulty of training resulted from the sparse reward signal in the large search space, which caused model probabilities for programs with non-zero reward to be very small at the beginning. If the beam size $k$ is small, good programs fall off the beam, leading to zero gradients for all programs in the beam. If the beam size $k$ is large, training is very slow, and the normalized probabilities of good programs when the model is untrained are still very small, leading to (1) near zero baselines, thus near zero gradients on "bad" programs (2) near zero gradients on good programs due to the low probability $P_\theta(a_{0:T} \mid x)$. To combat this, we present an alternative training strategy based on maximum-likelihood.

**Iterative ML**  If we had gold programs, we could directly optimize their likelihood. Since we do not have gold programs, we can perform an iterative procedure (similar to hard EM), where we search for good programs given fixed parameters, and then optimize the probability of the best program found so far. We do decoding on an example with a large beam size and declare $a_{0:T}^{best}(x)$ to be the pseudo-gold program that achieved highest reward with shortest length among the programs decoded on $x$ in all previous iterations. Then, we can optimize the ML objective:

$$J^{ML}(\theta) = \sum_x \log P_\theta(a_{0:T}^{best}(x) \mid x) \qquad (1)$$

A question $x$ is not included if we did not find any program with positive reward.

Training with iterative ML is fast because there is at most one program per example and the gradient is not weighted by model probability. while decoding with a large beam size is slow, we could train for multiple epochs after each decoding. This iterative process has a bootstrapping effect that a better model leads to a better program $a_{0:T}^{best}(x)$ through decoding, and a better program $a_{0:T}^{best}(x)$ leads to a better model through training.

Even with a large beam size, some programs are hard to find because of the large search space. A common solution to this problem is to use curriculum learning (Zaremba and Sutskever, 2015; Reed and de Freitas, 2016). The size of the search space is controlled by both the set of functions used in the program and the program length. We apply curriculum learning by gradually increasing both these quantities (see details in Section 3) when performing iterative ML.

Nevertheless, iterative ML uses only pseudo-gold programs and does not optimize the objective we truly care about. This has two adverse effects: (1) The best program $a_{0:T}^{best}(x)$ could be a spurious program that accidentally produces the correct answer (e.g., using the property PLACEOF-BIRTH instead of PLACEOFDEATH when the two places are the same), and thus does not generalize to other questions. (2) Because training does not observe full negative programs, the model often fails to distinguish between tokens that are related to one another. For example, differentiating PARENTSOF vs. SIBLINGSOF vs. CHILDRENOF can be challenging. We now present learning where we combine iterative ML with REINFORCE.

**Augmented REINFORCE**  To bootstrap REINFORCE, we can use iterative ML to find pseudo-gold programs, and then add these programs to the beam with a reasonably large probability. This is similar to methods from imitation learning (Ross et al., 2011; Jiang et al., 2012) that define a proposal distribution by linearly interpolating the model distribution and an oracle.

Algorithm 1 describes our overall training procedure. We first run iterative ML for $N_{ML}$ iterations and record the best program found for every example $x_i$. Then, we run REINFORCE, where we normalize the probabilities of the programs in beam to sum to $(1 - \alpha)$ and add $\alpha$ to the probability of the best found program $C^*(x_i)$.

**Algorithm 1** IML-REINFORCE

---

**Input:** question-answer pairs $\mathbb{D} = \{(x_i, y_i)\}$, mix ratio $\alpha$, reward function $R(\cdot)$, training iterations $N_{ML}$, $N_{RL}$, and beam sizes $B_{ML}$, $B_{RL}$.
**Procedure:**
Initialize $C_x^* = \phi$ the best program so far for $x$
Initialize model $\theta$ randomly ▷ Iterative ML
**for** $n = 1$ to $N_{ML}$ **do**
 **for** $(x, y)$ in $D$ **do**
 $\mathbb{C} \leftarrow$ Decode $B_{ML}$ programs given $x$
 **for** $j$ in $1...|\mathbb{C}|$ **do**
 **if** $R_{x,y}(C_j) > R_{x,y}(C_x^*)$ **then** $C_x^* \leftarrow C_j$
 $\theta \leftarrow$ ML training with $\mathbb{D}_{ML} = \{(x, C_x^*)\}$
Initialize model $\theta$ randomly ▷ REINFORCE
**for** $n = 1$ to $N_{RL}$ **do**
 $\mathbb{D}_{RL} \leftarrow \emptyset$ is the RL training set
 **for** $(x, y)$ in $D$ **do**
 $\mathbb{C} \leftarrow$ Decode $B_{RL}$ programs from $x$
 **for** $j$ in $1...|\mathbb{C}|$ **do**
 **if** $R_{x,y}(C_j) > R_{x,y}(C_x^*)$ **then** $C_x^* \leftarrow C_j$
 $\mathbb{C} \leftarrow \mathbb{C} \cup \{C_x^*\}$
 **for** $j$ in $1...|\mathbb{C}|$ **do**
 $\hat{p}_j \leftarrow (1-\alpha) \cdot \frac{p_j}{\sum_{j'} p_{j'}}$ where $p_j = P_\theta(C_j \mid x)$
 **if** $C_j = C_x^*$ **then** $\hat{p}_j \leftarrow \hat{p}_j + \alpha$
 $\mathbb{D}_{RL} \leftarrow \mathbb{D}_{RL} \cup \{(x, C_j, \hat{p}_j)\}$
 $\theta \leftarrow$ REINFORCE training with $\mathbb{D}_{RL}$

---

Consequently, the model always puts a reasonable amount of probability on a program with high reward during training.

On top of imitation learning, our approach is related to the common practice in reinforcement learning (Schaul et al., 2016) to replay rare successful experiences to reduce the training variance and improve training efficiency. This is also similar to recent developments (Wu et al., 2016) in machine translation, where ML and RL objectives are linearly combined, because anchoring the model to some high-reward outputs stabilizes training.

## 3 Experiments and Analysis

We now empirically show that NSM can learn a semantic parser from weak supervision over a large KB. We evaluate on WEBQUESTIONSSP, a challenging semantic parsing dataset with strong baselines. Experiments show that NSM achieves new state-of-the-art performance on WEBQUESTIONSSP with weak supervision, and significantly closing the gap between weak and full supervisions for this task. Hyper-parameter settings are in the supplemental material due to page limits.

### 3.1 The WEBQUESTIONSSP dataset

Collecting semantic parse labels is a difficult, time consuming task, but can increase the accuracy of state-of-the-art question-answering systems by full supervision. The WEBQUESTIONSSP dataset (Yih et al., 2016) contains semantic parses for the questions from WEBQUESTIONS (Berant et al., 2013) that are answerable using Freebase. It consists of 3,098 question-answer pairs for training and 1,639 for testing. These questions were collected using Google Suggest API and the answers were originally obtained using Amazon Mechanical Turk and updated by annotators who were familiar with the design of Freebase. We further separated out 620 questions from the training set as a validation set. For query pre-processing we used an in-house named entity linking system to find the entities in a question. The quality of the entity linker is similar to that of (Yih et al., 2015) at $94\%$ of the gold root entities being included. Similar to Dong and Lapata (2016), we replaced named entity tokens with a special token "ENT". For example, the question *"who plays meg in family guy"* is changed to *"who plays ENT in ENT ENT"*.

Following (Yih et al., 2015) we used the last publicly available snapshot of Freebase (Bollacker et al., 2008). Since NSM training requires random access to Freebase during decoding, we pre-processed Freebase by removing predicates that are not related to world knowledge (starting with *"/common/"*, *"/type/"*, *"/freebase/"*),[2] and removing all text valued predicates, which are rarely the answer. Out of all 23K relations, 434 relations are removed during preprocessing. This results in a graph that fits in memory with 23K relations, 82M nodes, and 417M edges.

### 3.2 Model And Training Details

For pre-trained word embeddings, we used the 300 dimension GloVe word embeddings trained on 840B tokens (Pennington et al., 2014). On the encoder side, we added a projection matrix to transform the embeddings into 50 dimensions. On the decoder side, we used the same GloVe embeddings to construct an embedding for each property using its Freebase id, and also added a projection matrix to transform this embedding to 50 dimension. A Freebase id contains three parts: domain, type, and property. For example, the Freebase id for PARENTSOF is *"/people/person/parents"*. *"people"* is the domain, *"person"* is the type and *"parents"* is the property. The embedding is constructed by concatenating the average of word embeddings in the domain and type name

---

[2] We kept "/common/topic/notable_types".

to the average of word embeddings in the property name. For example, if the embedding dimension is 300, the embedding dimension for *"/people/person/parents"* will be 600. The first 300 dimensions will be the average of the embeddings for *"people"* and *"person"*, and the second 300 dimensions will be the embedding for *"parents"*.

Inspired by the staged generation process in Yih et al. (2015), curriculum learning includes two steps. We first run iterative ML for 10 iterations with programs constrained to only use the "Hop" function and the maximum number of expressions is 2. Then, we run iterative ML again, but use both "Hop" and "Filter", and the maximum number of expressions is 3. However, the relations used by "Hop" are restricted to those that appeared in $a_{0:T}^{best}(q)$ in the first step.

### 3.3 Results and discussion

We evaluate performance using the offical evaluation script for WEBQUESTIONSSP. Because the answer to a question may contain multiple entities or values, precision, recall and F1 are computed based on the output of each individual question, and average F1 is reported as the main evaluation metric. Accuracy measures the proportion of questions that are answered exactly.

A comparison to STAGG, the prior state-of-the-art model (Yih et al., 2016, 2015) is shown in Table 2. Our model beats STAGG with weak supervision by a significant margin on all metrics, while relying on no feature engineering or hand-crafted rules. When STAGG is trained with strong supervision it obtains an F1 of 71.7, and thus NSM closes half the gap between training with weak and strong supervision.

| Model | Prec. | Rec. | F1 | Acc. |
|-------|-------|------|------|------|
| *STAGG* | 67.3 | 73.1 | 66.8 | 58.8 |
| *NSM* | 70.8 | 76.0 | **69.0** | 59.5 |

Table 2: Results on the test set. Average F1 is the main evaluation metric and NSM outperforms STAGG with no domain-specific knowledge or feature engineering.

Four key ingredients lead to the final performance of NSM. The first one is the neural computer interface that provides code assistance by checking for syntax and semantic errors. We find that semantic checks are very effective for open-domain KBs with a large number of properties. For our task, the average number of choices is reduced from 23K per step (all properties) to less

than 100 (the average number of relations connected to an entity).

The second ingredient is augmented REINFORCE training. Table 3 compares augmented REINFORCE, REINFORCE, and iterative ML on the validation set. REINFORCE gets stuck in local maxima and performs poorly, while iterative ML training is not directly optimizing the F1 measure, and achieves sub-optimal results. In contrast, augmented REINFORCE is able to bootstrap using pseudo-gold programs found by iterative ML and achieves the best performance on both the training and validation set.

| Settings | Train F1 | Valid F1 |
|----------|----------|----------|
| *iterative ML* | 68.6 | 60.1 |
| *REINFORCE* | 55.1 | 47.8 |
| *Augmented REINFORCE* | 83.0 | **67.2** |

Table 3: Average F1 on the validation set for augmented REINFORCE, REINFORCE, and iterative ML.

The third ingredient is curriculum learning during iterative ML. We compare the performance of the best programs found with and without curriculum learning in Table 4. We find that the best programs found with curriculum learning are substantially better than those found without curriculum learning by a large margin on every metric.

| Settings | Prec. | Rec. | F1 | Acc. |
|----------|-------|------|------|------|
| *No curriculum* | 79.1 | 91.1 | 78.5 | 67.2 |
| *Curriculum* | 88.6 | 96.1 | 89.5 | 79.8 |

Table 4: Evaluation of the program with the highest F1 score in the beam ($a_{0:t}^{best}$) with and without curriculum learning.

The last important ingredient is reducing overfitting. Given the small size of the dataset, overfitting is a major problem for training neural network models. We show the contributions of different techniques for controlling overfitting in Table 5. Note that after all the techniques have been applied, the model is still overfitting with training F1@1=83.0% and validation F1@1=67.2%.

**Error analysis** Two main sources of errors arise in manual error analysis

1. **Search failure**: the correct program is not found during search for pseudo-gold programs, either because the beam size is not large enough, or because the set of functions implemented by the interpreter is insufficient.

| Settings | $\Delta$ F1@1 |
|---|---|
| *-Pretrained word embeddings* | -5.5 |
| *-Pretrained property embeddings* | -2.7 |
| *-Dropout on GRU input and output* | -2.4 |
| *-Dropout on softmax* | -1.1 |
| *-Anonymize entity tokens* | -2.0 |

Table 5: Contributions of different overfitting techniques on the validation set.

The 89.5% F1 score in Table 4 indicates that at least 10% of the queries are of this kind.

2. **Ranking failure**: Pseudo-gold programs with high reward are found, but are not ranked at the top. Because training error is low, this is largely due to over-fitting. The 67.2% F1 score in Table 3 indicates that about 20% of examples are of this kind.

## 4 Related work

Among deep learning models for program induction, Reinforcement Learning Neural Turing Machines (RL-NTMs) (Zaremba and Sutskever, 2015) are the most similar to NSM, as a non-differentiable machine is controlled by a sequence model. Therefore, both models rely on REINFORCE for training. The main difference between the two is the abstraction level of the programming language. RL-NTM uses lower level operations such as memory address manipulation and byte reading/writing, while NSM uses a high level programming language over a large knowledge base that includes operations such as following properties from entities, or sorting based on a property, which is more suitable for representing semantics. Earlier work such as OOPS (Schmidhuber, 2004) has desirable characteristics, such as the ability to define new functions and modify code storage. These remain to be future improvements for NSM.

We formulate NSM training as an instance of reinforcement learning (Sutton and Barto, 1998) in order to directly optimize the task reward of the structured prediction problem (Norouzi et al., 2016). Compared to imitation learning methods (Daume et al., 2009; Ross et al., 2011) that interpolate a model distribution with an oracle, NSM needs to solve a challenging search problem of training from weak supervisions in a large program space. Our solution employs two techniques (a) a symbolic "computer" helps find good programs by pruning the search space (b) an it-

erative maximum likelihood (ML) process, where beam search is used to find pseudo-gold programs. Wiseman and Rush (Wiseman and Rush, 2016) proposed a max-margin approach to train a sequence-to-sequence scorer. However, their training procedure is more involved, and we did not implement it in this work.

NSM is similar to Neural Programmer (Neelakantan et al., 2015) and Dynamic Neural Module Network (Andreas et al., 2016) in that they all solve the problem of semantic parsing from structured data, and generate programs using similar semantics. The main difference between these approaches is how an intermediate result (the memory) is represented. Neural Programmer and Dynamic-NMN chose to represent results as vectors of weights (row selectors and attention vectors), which enables backpropagation and search through all possible programs in parallel. However, their strategy is not applicable to a large KB such as Freebase, which contains about 100M entities, and more than 20k properties. Instead, NSM chooses a more scalable approach, where the "computer" saves intermediate results, and the neural net only refers to them with variable names (e.g., "R1" for all cities in the US).

## 5 Conclusion

We propose the Manager-Programmer-Computer framework for neural program induction. It integrates neural networks with a symbolic *non-differentiable* computer to support *abstract*, *scalable* and *precise* operations through a friendly *neural computer interface*. Within this framework, we introduce the Neural Symbolic Machine, which integrates a sequence-to-sequence neural "programmer" with key-variable memory, and a Lisp interpreter with code assistance. Because the interpreter is non-differentiable and to directly optimize the task reward, we apply reinforcement learning and use pseudo-gold programs found by an iterative ML training process to bootstrap training. NSM achieves new state-of-the-art results on a challenging semantic parsing dataset with weak supervision, and significantly closing the gap between weak and full supervision. It is trained end-to-end, and does not require any feature engineering or domain-specific knowledge.

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

# A Supplemental Material

## A.1 More Model Details

The dimension of encoder hidden state, decoder hidden state and key embeddings are all 50. The embeddings for the functions and special tokens (e.g., "UNK", "GO") are randomly initialized by a truncated normal distribution with mean=0.0 and stddev=0.1. All the weight matrices are initialized with a uniform distribution in $[-\frac{\sqrt{3}}{d}, \frac{\sqrt{3}}{d}]$ where $d$ is the input dimension.

Dropout rate is set to 0.5, and we see a clear tendency that larger dropout produces better performance, indicating overfitting is a major problem for learning.

## A.2 More Training Details

In iterative maximum likelihood training, the decoding uses beam size $k = 100$ to update the approximate gold programs and the model is trained for 20 epochs after each decoding. We use Adam optimizer (Kingma and Ba, 2014) with initial learning rate 0.001 for optimization. In our experiment, this process usually converges after a few (5-8) iterations.

For REINFORCE training, the best hyperparameters are chosen using the validation set. We use beam size $k = 5$ for decoding, and $\alpha$ is set to 0.1. Because the dataset is small and some relations are only used once in the whole training set, we train the model on the entire training set for 200 iterations with the best hyperparameters. Then we train the model with learning rate decay until convergence. Learning rate is decayed as $g_t = g_0 \times \beta^{\frac{\max(0, t-t_s)}{m}}$, where $g_0 = 0.001$, $\beta = 0.5$ $m = 1000$, and $t_s$ is the number of training steps at the end of iteration 200.

Since decoding needs to query the knowledge graph constantly, the speed bottleneck for training is decoding. We address this problem in our implementation by partitioning the dataset, and using multiple decoders in parallel to handle each partition. We use 100 decoders, which queries 50

KG servers, and one trainer. The neural network model is implemented in TensorFlow. Since the model is small, we didn't see a significant speedup by using GPU, so all the decoders and the trainer are using CPU only.

## A.3  Extra Figures

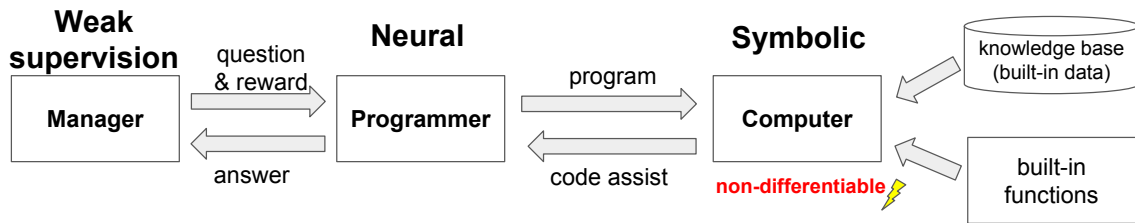

Figure 3:  Manager-Programmer-Computer framework

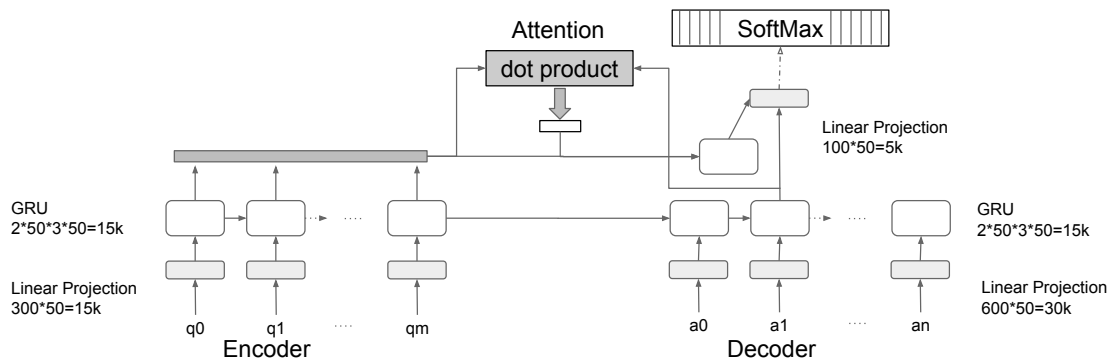

Figure 4:  Seq2Seq model architecture with dot-product attention and Dropout at GRU input, output, and softmax layers.

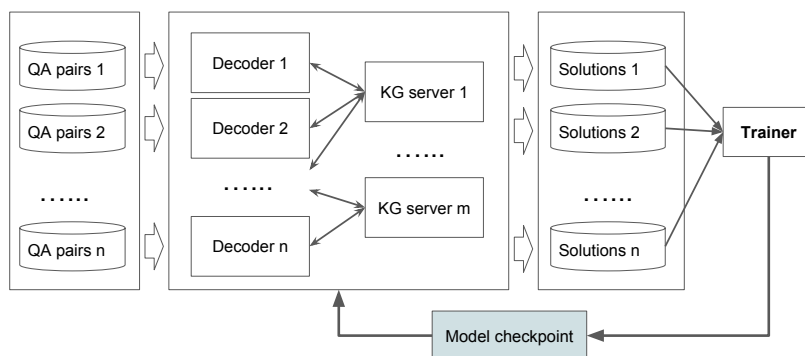

Figure 5:  System Architecture. 100 decoders, 50 KG servers and 1 trainer.

