# Peer review of "Neural Symbolic Machines: Learning Semantic Parsers on Freebase with Weak Supervision"

_ACL 2017 — decision unknown_

[Official Review · Reviewer 1 · rating 5 · confidence 4]
soundness 3 · originality 4 · clarity 5 · impact 3 · substance 4 · appropriateness 5 · meaningful comparison 3 · presentation format Oral Presentation

This paper introduces a new approach to semantic parsing in which the model is
equipped with a neural sequence to sequence (seq2seq) model (referred to as the
“programmer”) which encodes a natural language question and produces a
program. The programmer is also equipped with a ‘key variable’ memory
component which stores (a) entities in the questions (b) values of intermediate
variables formed during execution of intermediate programs. These variables are
referred to further build the program.                    The model is also equipped
with
certain
discrete operations (such as argmax or 'hop to next edges in a KB'). A separate
component ("interpreter/computer") executes these operations and stores
intermediate values (as explained before). Since the ‘programmer' is
inherently a seq2seq model, the "interpreter/computer” also acts as a
syntax/type checker only allowing the decoder to generate valid tokens. For
example, the second argument to the “hop” operation has to be a KB
predicate. Finally the model is trained with weak supervision and directly
optimizes the metric which is used to evaluate the performance (F score).
Because of the discrete operations and the non differentiable reward functions,
the model is trained with policy gradients (REINFORCE). Since gradients
obtained through REINFORCE have high variance, it is common to first pretrain
the model with a max-likelihood objective or find some good sequences of
actions trained through some auxiliary objective. This paper takes a latter
approach in which it finds good sequences via an iterative maximum likelihood
approach. The results and discussion sections are presented in a very nice way
and the model achieves SOTA results on the WebQuestions dataset when compared
to other weakly supervised model.

The paper is written clearly and is very easy to follow.

This paper presents a new and exciting direction and there is scope for a lot
of future research in this direction. I would definitely love to see this
presented in the conference.

Questions for the authors (important ones first)

1. Another alternative way of training the model would be to bootstrap the
parameters (\theta) from the iterative ML method instead of adding pseudo gold
programs in the beam (Line 510 would be deleted). Did you try that and if so
why do you think it didn’t work?
2. What was the baseline model in REINFORCE. Did you have a separate network
which predicts the value function. This must be discussed in the paper in
detail.
3. Were there programs which required multiple hop operations? Or were they
limited to single hops. If there were, can you provide an example? (I will
understand if you are bound by word limit of the response)
4. Can you give an example where the filter operation would be used?
5. I did not follow the motivation behind replacing the entities in the
question with special ENT symbol

Minor comments:
Line 161 describe -> describing
Line 318 decoder reads ‘)’ -> decoder generates ‘)'

[Official Review · Reviewer 2 · rating 4 · confidence 4]
soundness 3 · originality 4 · clarity 4 · impact 3 · substance 4 · appropriateness 5 · meaningful comparison 3 · presentation format Oral Presentation

This paper introduces Neural Symbolic Machines (NSMs) --- a deep neural model
equipped with discrete memory to facilitate symbolic execution. An NSM includes
three components: (1) a manager that provides weak supervision for learning,
(2) a differentiable programmer based on neural sequence to sequence model,
which encodes input instructions and predicts simplified Lisp programs using
partial execution results stored in external discrete memories. (3) a symbolic
computer that executes programs and provide code assistance to the programmer
to prune search space. The authors conduct experiments on a semantic parsing
task (WebQuestionsSP), and show that (1) NSM is able to model language
compositionality by saving and reusing intermediate execution results, (2)
Augmented REINFORCE is superior than vanilla REINFROCE for sequence prediction
problems, and (3) NSM trained end-to-end with weak supervision is able to
outperform existing sate-of-the-art method (STAGG).

- Strengths

* The idea of using discrete, symbolic memories for neural execution models is
novel.                    Although in implementation it may simply reduce to copying
previously
executed variable tokens from an extra buffer, this approach is still
impressive since it works well for a large-scale semantic parsing task.

* The proposed revised REINFORCE training schema using imperfect hypotheses
derived from maximum likelihood training is interesting and effective, and
could inspire future exploration in mixing ML/RL training for neural
sequence-to-sequence models.

* The scale of experiments is larger than any previous works in modeling neural
execution and program induction. The results are impressive.

* The paper is generally clear and well-written, although there are some points
which might require further clarification (e.g., how do the keys ($v_i$'s in
Fig. 2) of variable tokens involved in computing action probabilities?
Conflicting notations: $v$ is used to refer to variables in Tab. 1 and memory
keys in Fig 1.).

Overall, I like this paper and would like to see it in the conference.

* Weaknesses

* [Choice of Dataset] The authors use WebQuestionsSP as the testbed. Why not
using the most popular WebQuestions (Berant et al., 2013) benchmark set? Since
NSM only requires weak supervision, using WebQuestions would be more intuitive
and straightforward, plus it could facilitate direct comparison with
main-stream QA research.

* [Analysis of Compositionality] One of the contribution of this work is the
usage of symbolic intermediate execution results to facilitate modeling
language compositionality. One interesting question is how well questions with
various compositional depth are handled. Simple one-hop questions are the
easiest to solve, while complex multi-hop ones that require filtering and
superlative operations (argmax/min) would be highly non-trivial. The authors
should present detailed analysis regarding the performance on question sets
with different compositional depth.

* [Missing References] I find some relevant papers in this field missing. For
example, the authors should cite previous RL-based methods for knowledge-based
semantic parsing (e.g., Berant and Liang., 2015), the sequence level REINFORCE
training method of (Ranzato et al., 2016) which is closely related to augmented
REINFORCE, and the neural enquirer work (Yin et al., 2016) which uses
continuous differentiable memories for modeling neural execution.

* Misc.

* Why is the REINFORCE algorithm randomly initialized (Algo. 1) instead of
using parameters pre-trained with iterative ML?

* What is KG server in Figure 5?